# Body shape and performance on the US Army Combat Fitness Test: Insights from a 3D body image scanner

**Maria Smith[1], Dusty Turner[2], Charlotte Spencer[1], Nicholas Gist[3], Sarah Ferreira[3], Kevin Quigley[1], Tyson Walsh[1], Nicholas Clark[1], William Boldt[1], Justin Espe[1], Diana M. Thomas[1]** *

**1** Department of Mathematical Sciences, United States Military Academy, West Point, New York, United States of America, **2** Center for Army Analysis, Fort Belvoir, Virginia, United States of America, **3** Department of Physical Education, United States Military Academy, West Point, New York, United States of America

* diana.thomas@westpoint.edu

## Abstract

### Objective

To identify relationships between body shape, body composition, sex and performance on the new US Army Combat Fitness Test (ACFT).

### Methods

Two hundred and thirty-nine United States Military Academy cadets took the ACFT between February and April of 2021. The cadets were imaged with a Styku 3D scanner that measured circumferences at 20 locations on the body. A correlation analysis was conducted between body site measurements and ACFT event performance and evaluated using Pearson correlation coefficients and p-values. A k-means cluster analysis was performed over the circumference data and ACFT performance were evaluated between clusters using t-tests with a Holm-Bonferroni correction.

### Results

The cluster analysis resulted in 5 groups: 1. "V" shaped males, 2. larger males, 3. inverted "V" shaped males and females, 4. "V" shaped smaller males and females, and 5. smallest males and females. ACFT performance was the highest in Clusters 1 and 2 on all events except the 2-mile run. Clusters 3 and 4 had no statistically significant differences in performance but both clusters performed better than Cluster 5.

### Conclusions

The association between ACFT performance and body shape is more detailed and informative than considering performance solely by sex (males and females). These associations may provide novel ways to design training programs from baseline shape measurements.

**Data Availability Statement:** Due to military security concerns, data on the cadet population at the United States Military Academy cannot be

publicly released without a review by the US Military Academy. Data will be made available by request after review by the United States Military Academy Chief Data Officer, Paul Evangelista. The review will determine the risks to personnel for data sharing and is dependent on the type of request. Requests can be made at paul. evangelista@westpoint.edu. Statistical analyses that were performed are available as R Markdown files in the supplementary online information.

**Funding:** Nicholas Clark, Diana M. Thomas and Nicholas Gist were was supported by National Institutes of Health (NIH) Inter Agency Agreement AOD22022001. The NIH had no role in study design, data collection and analysis, decision to publish, or preparation of the manuscript.

**Competing interests:** The authors have declared that no competing interests exist.

**Abbreviations:** ACFT, Army Combat Fitness Test; BMI, body mass index; DXA, dual-energy X-ray absorptiometry; FFMI, fat-free mass index; FMI, fat mass index; HRPU, hand release pushups; LT, leg tuck; MDL, 3-repitition maximum deadlift; SDC, sprint drag and carry; SPT, Standing Power Throw.

## Introduction

Based on a recent Research and Development (RAND) report [1], the United States Army has changed the new Army Combat Fitness Test (ACFT) from an age and sex independent physical assessment to an assessment that is scored normalized by age-and-sex-performance. In addition, one of the ACFT events, the leg tuck, is now replaced with a plank event. Prior to these changes, there existed substantial differences in performance between males and females on this event [2–4].

The original version of the ACFT was a six-event physical fitness test [2, 5] designed to be sex and age neutral that assesses whether soldiers are able to perform physical tasks required for combat, like casualty evacuation, movement under fire, construct a fighting position [2]. The concept behind age-sex neutrality was because all soldiers must accomplish these tasks, predicting performance from easily measurable individual factors and understanding what specific role sex plays is important for designing effective training programs.

Body size is one explanation for differences in performance between the sexes [6]. The body mass index (BMI) is routinely collected in Army soldiers [6, 7]. Higher BMI is associated with better performance on events requiring muscular strength and power such as medicine ball throws, sled drags, and dead lifts [8, 9]. On the other hand, higher BMI is inversely related to aerobic events such as the 2-mile run [8, 9]. BMI does not equate to body fat, however, it is known that high fat mass index (FMI) is associated with high fat free mass index (FFMI) [10], which in turn is associated with better performance on muscular strength activities. BMI also does not provide body shape insights which have been known to correlate to events such as pull-ups [11, 12].

We wanted to understand the role body shape on ACFT performance. Specifically, what are the archetype body shapes that predispose individuals to better ACFT performance and alternatively which archetype body shapes that are associated to poorer ACFT performance. Addressing these questions can offer opportunities for training programs to target specific body sites associated with better ACFT performance.

## Methods

### Study design

This was a cross-sectional observational study of 239 USMA cadets who took the ACFT between February and May of 2021. Relationships between 3D imaged body circumference-sand ACFT performance was examined.

Data collection utilized new technology that obtained anthropometric measurements quickly and efficiently for analysis. The cadets were scanned using 3D imaging technology which calculated anatomical circumferences at 20 locations on the body. These measurements were used to identify different body shape attributes related to performance on the specific ACFT events.

The study design has three main components. The first is data collection which involved collecting ACFT performance and body shape data. The second component focused on correlation between individual body shape parameters and their influence on performance. The final component characterized archetype body shapes identified in the data through a clustering algorithm and then compared performance over these clusters. Fig 1 is a flow diagram that describes the three stages of the study.

All statistical analyses were performed in the software package R (**R** Core Team (2013)). The R package "tidyr" [13] was used to perform summary statistics by grouping.

**Sample size estimates.** With significance level, $\alpha = 0.05$ (two-tailed), 80% power and an expected correlation of r = 0.20, the total sample size required to determine whether a

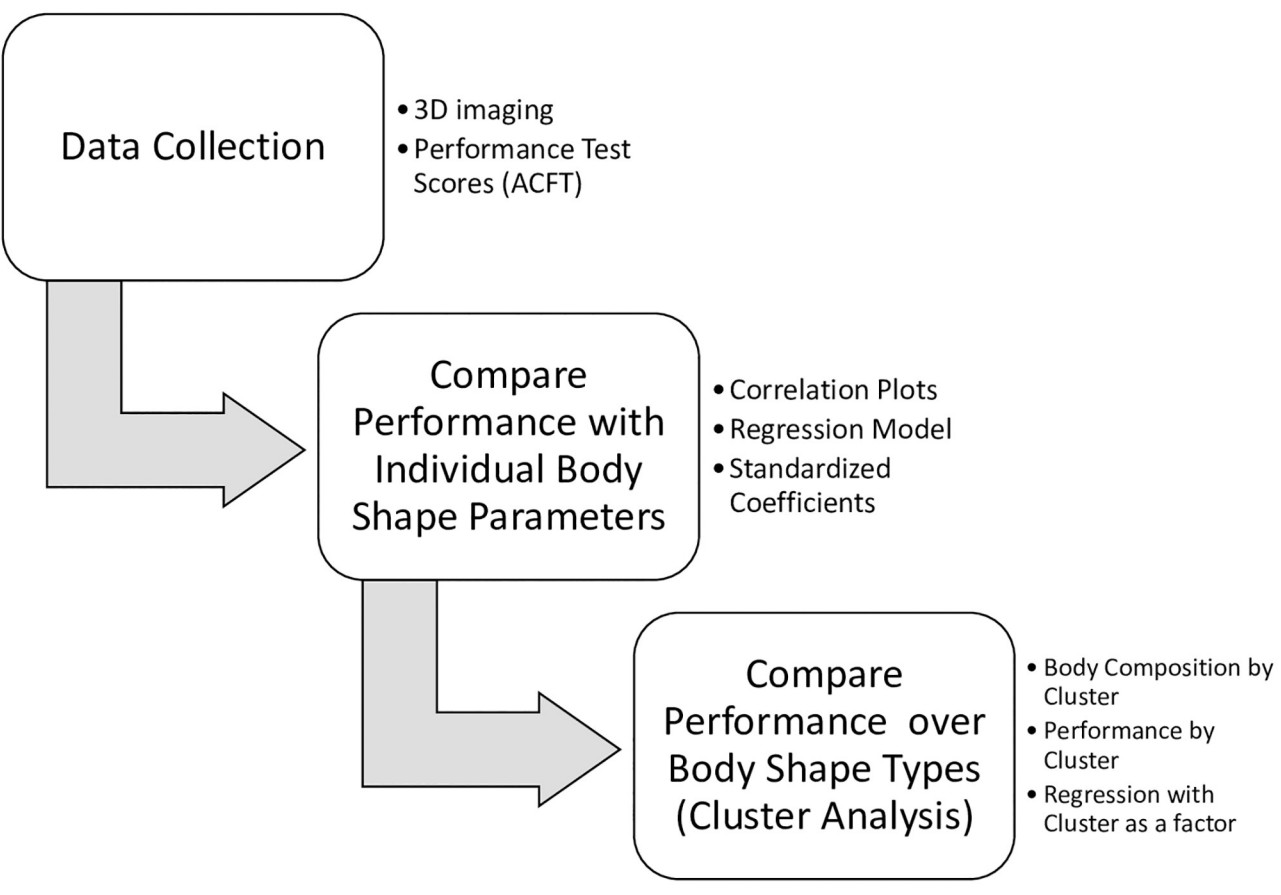

**Fig 1. Flowchart describing the study design.**

correlation coefficient differs from zero was 194 [14]. For multiple regression, we applied Green's rule [15] for 13 predictors. Setting significance level, $\alpha = 0.05$ (two-tailed), a medium effect (Cohen's f = 0.39), and 80% power, we arrived at a sample size of 154 to test the entire model and 117 to test the coefficients. Our convenience sample of 239 satisfied the sample size requirements.

**Study cohort.** Participants were recruited through posters, announcements in the dining facility, and emails disseminated by cadet strength and conditioning staff. Participants were eligible if they were currently USMA cadets who had taken or were going to take the ACFT between February and May of 2021. USMA cadets are between the ages of 18 to 29 years old. The study (Number CA20-009) was approved by the USMA Institutional Review Board and written informed consent was obtained from all study participants.

**The Army Combat Fitness Test (ACFT).** The six ACFT events [16] prior to March 2022 (Online S1 File), conducted in order with a maximum of 2 minutes between events, include a 3-Repetition Maximum Deadlift (MDL), Standing Power Throw (SPT), Hand-Release Push-Ups (HRPU), Sprint-Drag-Carry (SDC), Leg Tuck (LT), and the 2-mile run [17]. Graphic descriptions of the current ACFT that replaced LT with a plank can be found in the official ACFT website [18]. A image description of the LT can be found on the US Army Basic Training website [19].

The ACFT was taken by participants between the end of February 2021 and the beginning of May 2021, and scores were maintained by the USMA Office of Institutional Research office.

| Circumferences | Body Site (L=Left, R=Right) |
|---|---|
| A | Bicep (L, R) |
| B | Lower Bicep (L, R) |
| C | Forearm (L, R) |
| D | Upper Thigh (L, R) |
| E | Mid-thigh (L, R) |
| F | Lower Thigh (L, R) |
| G | Calf (L, R) |
| H | Chest |
| I | Abdominal Waist |
| J | Narrowest Waist |
| K | Lower Waist |
| L | High Hip |
| M | Hip |
| N | Neck |

**Fig 2.** The avatar (on right) and circumferences that are output from the Styku scan device software. Left and right circumference measurements were averaged for statistical analysis.

After participants were scanned, their raw scores on the ACFT were obtained from institutional research.

**3D body image scan measurements.** Between February and March of 2021, participants arrived at a designated location to be scanned using the Styku 3D body imaging machine (Styku S100, Advanced Model of Styku Phoenix software; Los Angeles, CA). Participants stood on a rotating turntable with their arms held in an "A" pose. As the turntable revolved for 35 seconds, a tower 114 cm away captured images using an infrared camera equipped with "Time of Flight" technology that quantifies the depth in an image [20]. Participants wore either their USMA issued swimsuit or compression shorts with a sports bra for female participants. The Styku internal software generates body circumference measurements at 20 different locations, ratios of chest-to-waist, chest-to-hip and waist-to-hip, and body weight. Styku has been validated against manual measurements for reliability [21]. All statistical analysis used averaged left and right limb circumferences (Fig 2), which reduced to 13 averaged circumferences.

## Statistical methods

**Correlation between body site measurements, percent body fat, and ACFT event performance.** We wished to evaluate how individual body circumference measured by the Styku

are associated with ACFT event-specific performance. To perform this analysis, we generated correlation plots, Pearson correlation coefficients, and p-values for each event score against each of the 13 circumferences, chest-to-waist ratio, chest-to-hip ratio, waist-to hip ratio, BMI, weight, and height, versus ACFT event-specific scores.

**Regression models predicting ACFT event specific performance.** Multiple linear regression models were developed using the 13 circumferences and ratios as independent variables and event performance as the dependent variable. Specifically, an individual multiple linear regression model was developed to predict ACFT performance scores; MDL, SPT, HRPU, SDC, LT, and the 2-mile run [17] from the Styku generated measurements.

Model coefficients in the final models were standardized and plotted with 95% confidence intervals to compare the contribution of each measured body site in the final model on performance.

**k-means cluster analysis.** To group the participant population by similar body shapes, a k-means cluster analysis was performed over the normalized body shape measurements. The normalized body shape measurements of the 13 circumference sites, body weight, and body height were computed by taking the difference of the measurement and the mean value. This difference was then divided by the standard deviation. As noted earlier, clustering is an unsupervised method which allows the data to define groupings and eliminates human preconceived groupings of populations (e.g. by sex). The number of clusters to retain was determined by the cluster number at the elbow of the scree plot. A heat map was generated by cluster identifying how far the body site locations were from the sample mean value.

**Regression models with cluster number as a covariate.** To see if cluster assignment improves prediction of performance on the ACFT events, we developed 6 multiple linear regression models with two independent variables, cluster number and sex. The explanatory variable was the performance score of the ACFT event.

## Results

### Participant characteristics

A total of 265 participants were enrolled in the study and scanned using the Styku. Of these, 239 completed the ACFT. In the reference dataset of 239 participants with both Styku measurements and ACFT scores,. In the reference dataset, there were 83 females and 156 males. Table 1 contains a summary of the data by sex, cluster number, and aggregate totals. for BMI, four of the 13 body circumference measurements (locations labeled in Fig 2), and ACFT fitness scores.

Included are BMI (kg/m$^2$), bicep, waist, thigh, chest, and forearm circumference. Females had lower BMI (22.36 ±2.22 m/kg$^2$) than males (25.33±2.99 m/kg$^2$) and smaller average circumferences than males.

**Body shape and ACFT performance.** *Correlation of body circumference and ACFT performance.* Table 2 contains the Pearson correlation coefficients between the measurements provided by the Styku against each ACFT event. The highest correlations over all events were held by bicep circumference (Label A Fig 2), chest-to-waist ratio, and forearm circumference (Label E Fig 2). The lowest correlations were observed in hip circumference, waist circumference, thigh circumference, and BMI.

BMI and circumferences sites at the waist and hip did not have statistically significant correlations to the 2-mile run (Online S2 File).

*Regression models predicting ACFT event specific performance.* Table 3 contains statistically significant model terms and the adjusted R$^2$ of models predicting ACFT event-specific scores body measurements (for full model details see Online S3 File). Bicep circumference appeared

**Table 1. Table contains mean (±SD) BMI and Styku body circumferences at five different sites by cluster, sex and totals.** Waist circumference was the Styku measurement at the abdomen (labeled "I" in Fig 1) and thigh circumference was the Styku measurement at the lower thigh (labeled "F" in Fig 1). Mean (±SD) of the raw scores for each ACFT event are also provided. Finally, mean (±SD) of percent body fat in 47 participants are presented by cluster with sample sizes: Cluster 1 (13) Cluster 2 (8), Cluster 3 (10), Cluster 4 (5), and Cluster 5 (11).

| Cluster Number & Group | 1 | 2 | 3 | 4 | 5 | Female | Male | Total |
|---|---|---|---|---|---|---|---|---|
| N | 70 | 27 | 50 | 39 | 53 | 83 | 156 | 239 |
| %F | 1 | 0 | 64 | 8 | 89 | - - - | - - - | 35 |
| BMI (kg/m$^2$) | 26.10±1.73 | 29.31±1.66 | 24.13±1.93 | 22.30±1.44 | 21.00±1.30 | 22.36±2.22 | 25.33±2.99 | 24.30±3.09 |
| %Fat* | 15.4±5.02 | 22.0±7.84 | 27.6±5.71 | 12.7±6.47 | 22.4±4.15 | 25.5±5.64 | 17.35±6.76 | 20.5±7.47 |
| Bicep (cm) | 32.65±1.92 | 34.95±2.24 | 28.16±2.16 | 28.67±2.02 | 24.37±1.51 | 25.60±2.18 | 31.55±3.20 | 29.48±4.05 |
| Waist (cm) | 78.57±3.10 | 88.19±3.92 | 77.49±4.24 | 71.92±3.45 | 68.89±3.65 | 72.31±5.41 | 78.27±6.56 | 76.20±6.80 |
| Thigh (cm) | 45.74±2.21 | 48.18±2.16 | 44.07±2.22 | 41.22±2.33 | 40.00±2.20 | 41.96±3.02 | 44.56±3.45 | 43.66±2.53 |
| Chest (cm) | 101.10±4.36 | 107.80±5.82 | 90.26±3.81 | 91.52±5.34 | 82.49±3.51 | 85.90±5.62 | 98.17±8.24 | 93.91±9.45 |
| Forearm (cm) | 28.21±1.61 | 29.48±1.63 | 24.79±2.09 | 25.32±1.72 | 22.00±1.20 | 22.81±1.77 | 27.37±2.36 | 25.79±3.07 |
| MDL (lb) | 331.30±19.26 | 324.10±33.54 | 225.00±43.81 | 272.60±44.94 | 178.10±26.24 | 186.60±30.49 | 306.20±43.59 | 264.70±69.38 |
| SPT (m) | 10.91±1.47 | 11.88±1.88 | 7.59±1.94 | 8.82±1.76 | 5.82±1.07 | 6.11±1.22 | 10.31±2.02 | 8.85±2.68 |
| HRPU | 51.04±8.10 | 45.00±10.02 | 34.34±10.70 | 46.85±10.50 | 32.60±10.21 | 31.25±9.73 | 47.86±9.55 | 42.09±12.44 |
| SDC (sec) | 87.33±5.98 | 89.22±8.08 | 108.60±14.07 | 97.90±11.83 | 123.11±12.30 | 119.50±12.80 | 92.17±11.13 | 101.60±17.50 |
| LT | 17.63±3.67 | 13.07±5.92 | 7.00±5.89 | 16.00±5.21 | 6.64±6.05 | 5.13±4.84 | 15.94±4.99 | 12.19±7.13 |
| 2-mile run (sec) | 854.10±83.74 | 916.90±123.60 | 962.10±122.2 | 880.90±128.70 | 957.30±113.90 | 963.90±15.60 | 880.30±110.80 | 911.10±119.90 |

*Body composition was collected in 47 of the participants with 40% females.

*Abbreviations are N—number of participants, %F—percent female, BMI—body mass index, MDL—maximum deadlift, SPT—standing power throw, HRPU—hand release push-up, SDC—sprint-drag-carry, LT—leg tuck.

*Body composition was collected in 47 of the participants with 40% females. The participant numbers for each cluster are: Cluster 1 (13) Cluster 2 (8), Cluster 3 (10), Cluster 4 (5), Cluster 5 (11)

**Table 2. Correlations between body height, site circumference, weight, BMI and % body fat with raw scores on the six ACFT events.** Significant correlations are denoted with asterisks at significance level *p < 0.05; **significant at p < 0.001; *** significant at p < 0.0001. Abbreviations are MDL—maximum deadlift, SPT—standing power throw, HRPU—hand-release push-up, SDC—spring drag carry, LT—leg tuck, BMI—body mass index.

| Measurement | MDL | SPT | HRPU | SDC | LT | 2-mile Run |
|---|---|---|---|---|---|---|
| Height | 0.66*** | 0.74*** | 0.31*** | -0.72*** | 0.42*** | -0.35*** |
| Bicep | 0.82*** | 0.78*** | 0.56*** | -0.72*** | 0.54*** | -0.23*** |
| Lower Bicep | 0.78*** | 0.77*** | 0.510*** | -0.72*** | 0.515*** | -0.22*** |
| Chest to Hip | 0.65*** | 0.55*** | 0.604*** | -0.54*** | 0.60*** | -0.27*** |
| Chest to Waist | 0.70*** | 0.58*** | 0.67*** | -0.61*** | 0.69*** | -0.35*** |
| Calf | 0.54*** | 0.53*** | 0.34*** | -0.52*** | 0.29*** | -0.10*** |
| Chest | 0.78*** | 0.75*** | 0.51*** | -0.69*** | 0.51*** | -0.21** |
| Forearm | 0.80*** | 0.79*** | 0.55*** | -0.74*** | 0.56*** | -0.23*** |
| High Hip | 0.32*** | 0.40*** | -0.03 | -0.33*** | -0.02 | 0.06 |
| Hip | 0.41*** | 0.47*** | 0.06 | -0.41*** | 0.06 | 0.01 |
| Mid-thigh | 0.65*** | 0.67*** | 0.31*** | -0.60*** | 0.26*** | -0.09 |
| Neck | 0.75*** | 0.71*** | 0.48*** | -0.65*** | 0.50*** | -0.28*** |
| Lower Thigh | 0.57*** | 0.58*** | 0.26*** | -0.53*** | 0.21*** | -0.11 |
| Upper Thigh | 0.58*** | 0.57*** | 0.21*** | -0.52*** | 0.20*** | -0.02 |
| Waist Abdominal | 0.55*** | 0.60*** | 0.19*** | -0.49*** | 0.15* | -0.03 |
| Waist to Hip | 0.42*** | 0.44*** | 0.26*** | -0.33*** | 0.19** | -0.08 |
| Lower Waist | 0.18** | 0.27*** | -0.14* | -0.17** | -0.18** | 0.15* |
| Narrowest Waist | 0.74*** | 0.75*** | 0.40*** | -0.66*** | 0.38*** | -0.17** |
| Weight | 0.76*** | 0.81*** | 0.37*** | -0.73*** | 0.38*** | -0.21** |
| BMI | 0.64*** | 0.63*** | 0.32*** | -0.54*** | 0.25*** | -0.04 |
| % Body Fat | -0.50*** | -0.50*** | -0.65*** | 0.67*** | -0.65*** | 0.59*** |

**Table 3. Statistically significant terms and adjusted R² for regression model predicting event scores from Styku measured circumferences.** Additional model details are provided in the Online S6 File.

|  | MDL | SPT | HRPU | SDC | LT | 2-mile run |
|---|---|---|---|---|---|---|
| **Adj R²** | 0.83 | 0.77 | 0.59 | 0.72 | 0.67 | 0.25 |
| **Statistically significant (p<0.05) terms** | Sex (p<0.001) | Sex (p = 0.005) | Sex (p = 0.002) | Sex (p = 0.022) | Sex (p<0.001) | Height (p<0.001) |
|  | Bicep (p = 0.003) | Height (0.005) | Bicep (p<0.001) | Height (p<0.001) | Bicep (p<0.001) | Neck (p = 0.002) |
|  |  | Forearm (0.014) | Upper Thigh (p = 0.034) | Chest (p = 0.032) | Forearm (p = 0.024) | Lower Thigh (p = 0.006) |
|  |  |  |  |  | Lower Bicep (p = 0.034) | Upper Thigh (p = 0.041) |

*Abbreviations are MDL—maximum deadlift, SPT—standing power throw, HRPU—hand release push-up, SDC—sprint-drag-carry, LT—leg tuck.

in the MDL, HRPU, and LT with coefficients indicating larger bicep circumferences was associated with better scores on these events. Height appeared as a statistically significant term in the SPR, SDC, and 2-mile run models with taller individuals associated with better scores on the SPR and 2-mile run, and lower scores on the SDC. Males were positively related to performance for all events except the 2-mile run. Fig 3 depicts the standardized coefficients and 95% confidence intervals for model terms. Larger lower bicep circumference was related to lower performance on all events while, higher chest-to-waist ratio was associated to positive performance on all events. Larger forearm circumference was associated to positive performance on all events except the 2-mile run.

**Cluster analysis.** Based the number of clusters at the elbow of the scree plot (Online S4 File), five clusters were retained for analysis. A cluster heat map appears in Fig 4 with darker shades of green representing measurements further below the mean and darker shades of red representing measurements higher than the mean. Cluster 1 consisted of primarily "V-shaped" males (only 1 female) with chest-to-waist ratio over 1. Cluster 2 contained only males that had the highest BMI and waist circumference over all clusters. Cluster 2 had a chest-to-waist ratio of 1.14 ± 0.6 while Cluster 1 had a chest-to-waist ration of 1.19 ± 0.06. The third cluster contains a combination of both males and females. The chest-to-hip ratio is 0.86 ± 0.4 yielding an average body shape in this cluster that is an inverted "V". Cluster 4 consists of a majority male population that were "V" shaped and had smaller than average body circumferences (Fig 5). Finally, Cluster 5 consists of primarily females with body circumferences below average and inverted "V" shapes. The Cluster 5 population had lower BMI (21.00 ±1.30) compared to Cluster 3 (24.13 ±1.93).

*ACFT performance by cluster.* Overall average performance on every ACFT event was highest in Cluster 1 on every event except the SPT (Table 1, Box Plots Online S3 File). Participants classified to Cluster 2 had the highest performance on the SPT event and second highest performance on the MDL and SDC events. Individuals in this cluster had similar performance to Cluster 4 on the HRPU event and performed below Cluster 4 on the LT and the 2-mile run. Participants in Cluster 3 performed above Cluster 5 on all events except the 2-mile run.

Comparison of means between clusters revealed that Clusters 1 and 2 had statistically significant better performance than Clusters 3, 4 and 5 for the MDL (Online S5 File). Additionally, Clusters 3 and 4 had significantly better performance on the MDL compared to Cluster 5. For SPT, HRPU and SDC and LT, Clusters 1 and 2 had statistically significant better performance than Clusters 3 and 5. Finally, Cluster 4 had statistically significant better performance on the SDC and LT compared to Cluster 5. There were no statistically significant differences in performance over any ACFT event between Cluster 3 and Cluster 4.

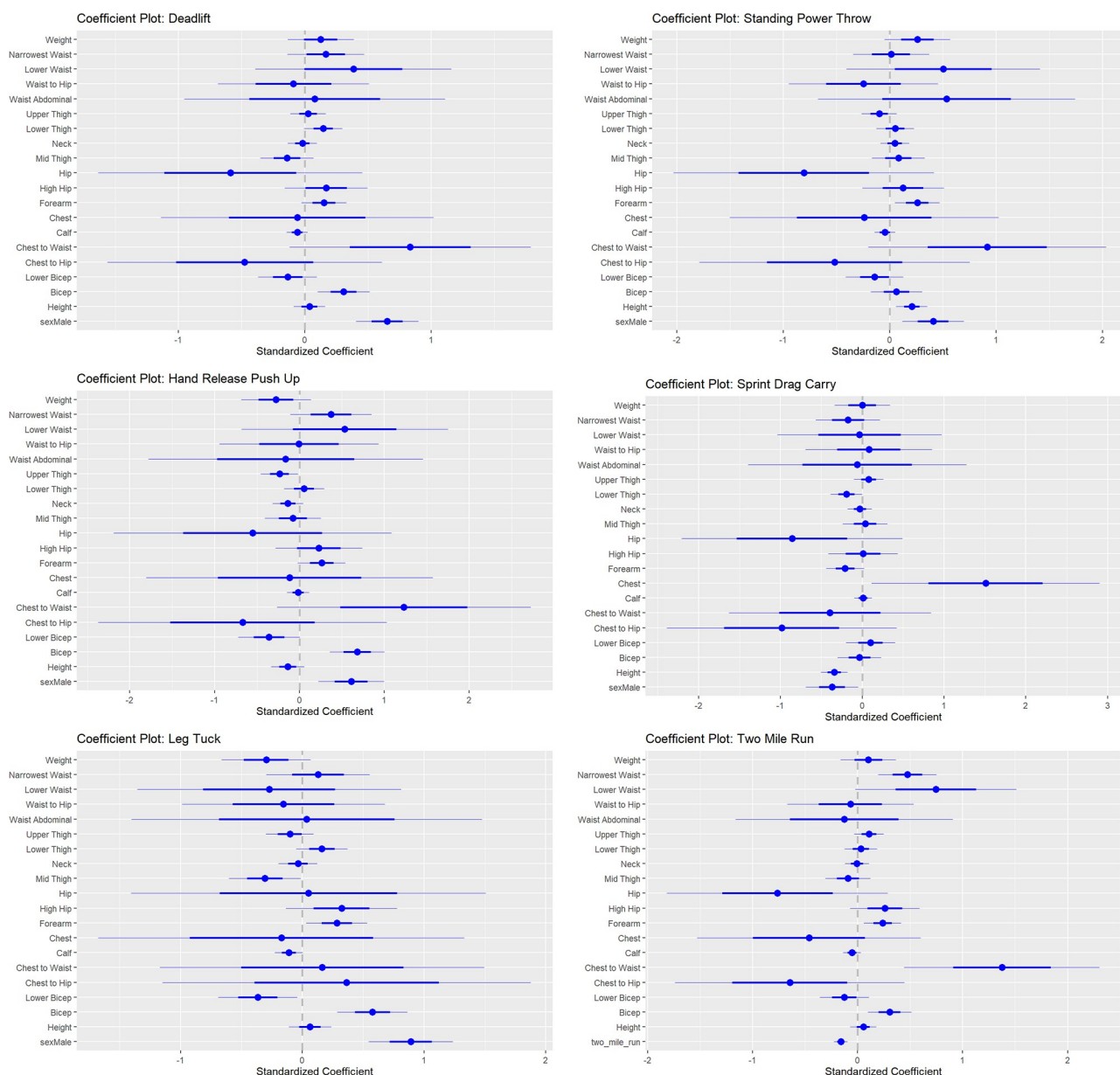

**Fig 3. Standardized coefficients resulting from regression models that predict performance by event from Styku measurements.** Positive coefficients represent terms that lead to higher scores, negative coefficients represent terms that lead to lower scores. Coefficients highest in magnitude are most influential.

*Regression models with cluster number as a covariate.* Similar adjusted $R^2$ were found for the six models predicting performance on each ACFT event compared to the original multiple linear regression models that were developed without cluster number as a covariate. The full statistical output can be found in Online S6 File as a R Markdown file.

## Discussion

There are differences between male and female performance on military fitness tests [4, 22–25]; however, grouping performance data solely by sex places males and females into distinct

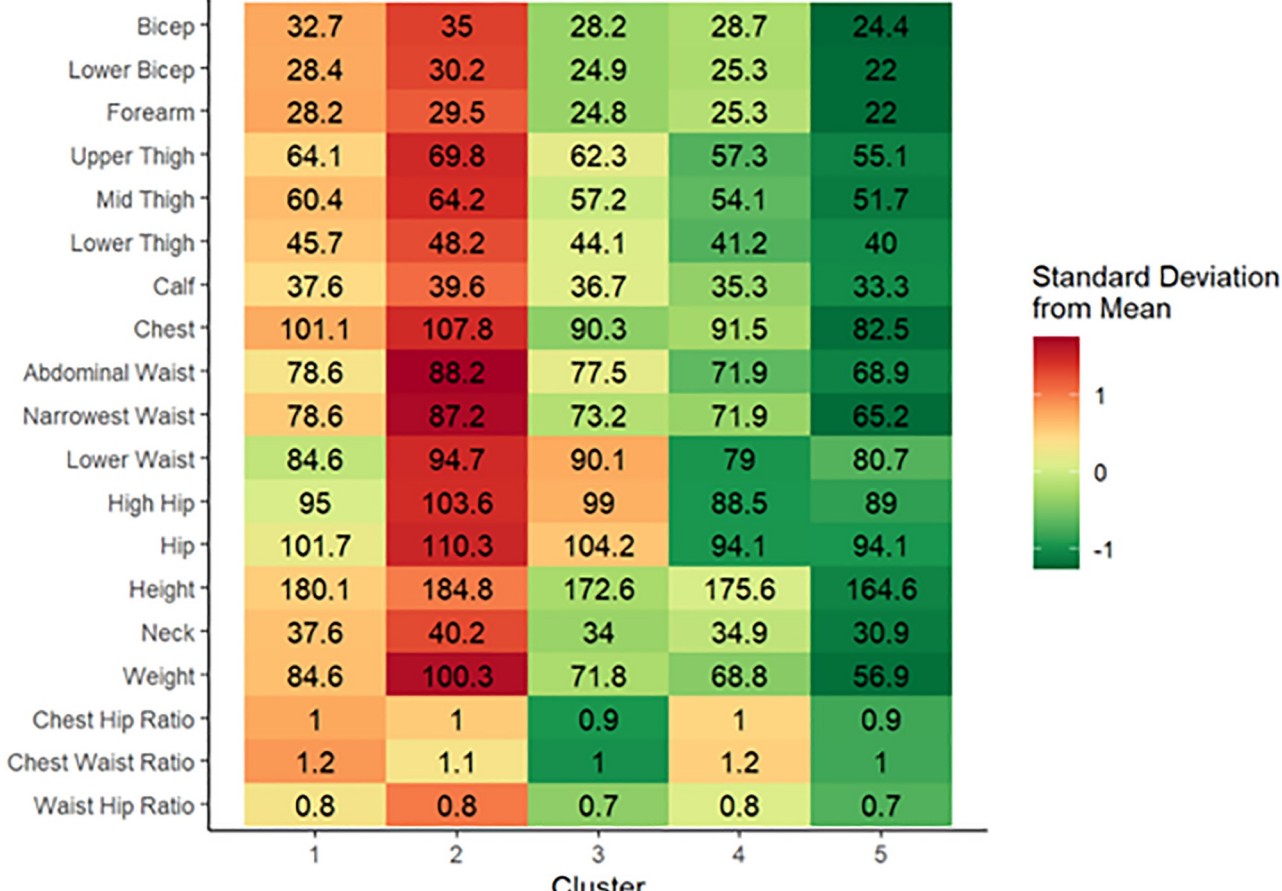

**Fig 4. Heat map of body measurements by cluster depicting the number of standard deviations from the mean value.** Red shading indicates that the value is above the mean with darker red shading corresponding to higher distance away from the mean. Similarly, green shading indicates that the value is below the mean with darker green corresponding to a higher distance away from the mean.

clusters with no overlap. Here we used a clustering algorithm that grouped 239 cadets at the United States Military Academy into five clusters of similar body shapes. Of these five groups, three resulted in a combination of males and females. While the top performing clusters were predominately male, the next performance rank was held by two clusters with a combination of males and females.

Existing studies in the US Army predicting physical performance largely focused on relationships between body composition [26–28] and the phased out Army Physical Fitness Test (APFT). Because the ACFT only became the US Army physical assessment of record in October 2022 [29], there is only one existing study on factors that predict performance. A recent study in 68 U.S. Army Reserve Officer Training Corps (ROTC) cadets measured body composition using bioimpedance and examined relationships between body composition and ACFT performance [22]. The study [22] found that there were significant correlations between fat free mass index (FFMI) and fat mass index (FMI) with overall ACFT performance. These findings are complementary with our cluster analysis results in Cluster 1 outperforming other archetype shapes on the ACFT. The study [22] did not find correlations between BMI and overall ACFT scores, which is in contrast to our results in Table 2. We found that BMI was correlated to positive performance on all ACFT events except the 2-mile run.

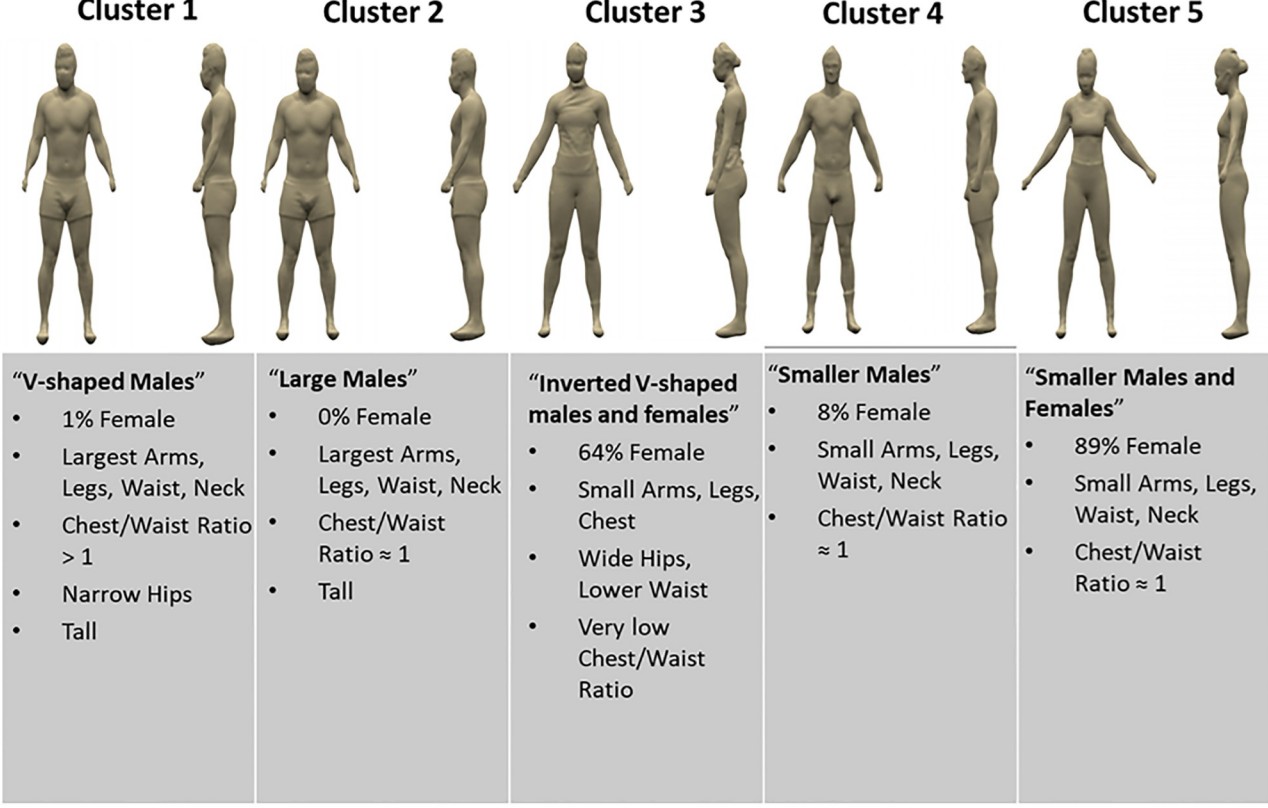

**Fig 5. Visual depictions and descriptions of body shape by cluster.**

The ACFT represents a combination of events that require muscular strength, power and anaerobic endurance (MDL, SPT, SDC) and aerobic endurance (2-mile run). To our knowledge, the role of body composition on the ACFT events like HRPU and LT have not been reported; however several studies have found that higher percent body fat is correlated to lower push-up and pull-up performance [11, 12, 30, 31].

The analysis here using 3D imaged body circumference also extends the literature by identifying key body circumferences that are associated with performance. Larger forearm and bicep circumference were associated with better performance on all events excluding the 2-mile run. On the other hand, larger lower bicep and mid-thigh circumferences were associated with lower performance on the HRPU, LT and 2-mile run. Because the circumference at the lower bicep in females was significantly correlated to percent body fat with $r = 0.70$, $p < 0.001$, the lower bicep may be a proxy for body fat in females.

## Sex-specific differences in body shape and ACFT performance

It is well known that athletic performance differs between males and females [32–34] and that this difference is physiological. For example, females have lower body mass, muscle mass, strength, lower bone mass, and lower oxygen-transport capacity than males [35–37] and

females in general have higher percent body fat and FMI than males [10, 38]. These findings were concordant with our study, because within each cluster, males in general outperformed females on the ACFT events. Moreover, larger bicep and forearm circumference were identified as important factors predicting performance on all events except the 2-mile run. The largest bicep and forearm circumferences were exhibited by Clusters 1 and 2 which were predominately male (only 1 female). However, previous research on trained body builders found that female bicep circumferences are smaller than males [39, 40]. This suggests that even with training, it may not be feasible for females to have comparable bicep circumferences to males.

Despite our concordant findings of sex-differences in performance, we note that Clusters 3 and 4 did not have statistically significant differences in ACFT performance, while both Clusters 3 and 4 did have statistically significant differences in performance compared to Cluster 5. Solely focusing on differences in physical performance between males and females would have obscured this finding. Instead designing training programs to shift Cluster 5 body shapes to Cluster 3 or 4 may be more encompassing. It remains to be shown if altering body shapes will improve performance.

## Study strengths and limitations

The use of the Styku 3D scanner, was a major strength of the study. The Styku has undergone validity and reliability testing compared to manual measurements [21]. Using the Styku allowed for numerous anthropometry measurements collected quickly and reliably in the cadet population at USMA. In addition, the carefully curated routine collection and storage of physical performance data at USMA allowed us to examine relationships between body shape and ACFT performance.

On the other hand, USMA cadets are not representative of the larger Army or the US population. In addition, the ACFT was not incorporated into USMA grades and cadets may not have been performing at maximal ability. However, our convenience sample may have self-selected motivated cadets.

There are several statistical limitations of our study. Human body proportions and circumferences have known relationships previously identified through allometric modeling [41, 42]. There may be implausible combinations of body circumferences that do not obey these relationships reflected by the regression models. Despite this, the standardized model coefficients provided insights that were congruent with the literature. Moreover, the cluster analysis was performed so that each participant's circumferences were one data array in the cluster analysis.

Finally, the identified relationships between body shape and performance is correlation which does not translate to causation. Longitudinal evidence is needed to determine whether changed body shape will lead to changes in physical performance on the ACFT. Moreover, with the new sex and age-specific scoring system and the replacement of the leg tuck event with a plank [1], some of our findings do not transfer to the modified ACFT.

Research on predicting performance has relied on body composition to predict performance [26]. Measurements from 3D body image scanners provide a new wealth of more detailed information on body shape that advances our understanding of which body types are associated with better performance. The scanners are affordable, do not require a special license to operate, provide measurements within seconds, and do not expose participants to radiation. Thus, they offer a feasible method to identify individuals and enroll them into targeted training programs.

## Conclusions

Body shape predicts performance on the new ACFT. Archetype body shape groups are associated with different levels of physical performance, providing new opportunities for targeted training that go beyond grouping populations by sex.

## Supporting information

**S1 File. The description of the events on the Army Combat Fitness Test.**
(DOCX)

**S2 File. Correlation matrix with distribution plots for correlations between body site measurement and performance on all 6 ACFT events.**
(PDF)

**S3 File. Box plots depicting ACFT performance by cluster.**
(PDF)

**S4 File. Scree plot depicting reduction in variation with the addition of each cluster.** The elbow of the scree plot occurs at 5 clusters.
(DOCX)

**S5 File. Pairwise comparison of difference in means by cluster on ACFT performance by event.**
(PDF)

**S6 File. R Markdown file with regression models predicting performance with cluster number as a covariate.**
(PDF)

**S7 File. R Markdown file with cluster analysis code.**
(PDF)

## Acknowledgments

The authors thank the following colleagues for reading and commenting on the final drafts of the manuscript prior to submission: Leslie Jones, Blake Schwartz, and Daniel Baller.

## Author Contributions

**Conceptualization:** Maria Smith, Nicholas Gist, Diana M. Thomas.

**Data curation:** Maria Smith, Charlotte Spencer, Nicholas Gist, Tyson Walsh, Justin Espe.

**Formal analysis:** Maria Smith, Dusty Turner, Charlotte Spencer, Kevin Quigley, Tyson Walsh, Nicholas Clark.

**Investigation:** Maria Smith, Charlotte Spencer, Sarah Ferreira, Tyson Walsh, William Boldt, Justin Espe.

**Methodology:** Maria Smith, Charlotte Spencer, Sarah Ferreira, Kevin Quigley, Tyson Walsh, Nicholas Clark, William Boldt, Justin Espe, Diana M. Thomas.

**Project administration:** Maria Smith, Diana M. Thomas.

**Supervision:** Nicholas Gist, Sarah Ferreira, Kevin Quigley, Diana M. Thomas.

**Visualization:** Maria Smith, Dusty Turner, Tyson Walsh.

Writing – **original draft:** Maria Smith, Charlotte Spencer, Diana M. Thomas.

Writing – **review & editing:** Maria Smith, Dusty Turner, Charlotte Spencer, Nicholas Gist, Sarah Ferreira, Kevin Quigley, Tyson Walsh, Nicholas Clark, William Boldt, Justin Espe, Diana M. Thomas.

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
