## [Decision Letter · Decision Letter 0]

28 Oct 2022

PONE-D-22-21891Body shape, body composition, and performance on the US Army Combat Fitness Test: Insights from a 3D body image scannerPLOS ONE

Dear Dr. Thomas,

Thank you for submitting your manuscript to PLOS ONE. After careful consideration, we feel that it has merit but does not fully meet PLOS ONE’s publication criteria as it currently stands. Therefore, we invite you to submit a revised version of the manuscript that addresses the points raised during the review process.

We finally managed to acquire the number of reviews needed. Please find all the comments below.==============================

We look forward to receiving your revised manuscript.

Kind regards,

Inge Roggen, M.D., Ph.D.

Academic Editor

PLOS ONE

Journal Requirements:

 “The senior author Diana M. Thomas was supported by NIH U54TR004279”

3.  Please expand the acronym “NIH” (as indicated in your financial disclosure) so that it states the name of your funders in full.

“No”

7. Please amend the manuscript submission data (via Edit Submission) to include author “Dusty Turner”

Reviewers' comments:

Reviewer's Responses to Questions

**Comments to the Author**

1. Is the manuscript technically sound, and do the data support the conclusions?

Reviewer #1: Yes

Reviewer #2: Yes

Reviewer #3: Yes

2. Has the statistical analysis been performed appropriately and rigorously? 

Reviewer #1: Yes

Reviewer #2: Yes

Reviewer #3: Yes

3. Have the authors made all data underlying the findings in their manuscript fully available?

Reviewer #1: Yes

Reviewer #2: Yes

Reviewer #3: No

4. Is the manuscript presented in an intelligible fashion and written in standard English?

Reviewer #1: Yes

Reviewer #2: Yes

Reviewer #3: Yes

5. Review Comments to the Author

Reviewer #1: Dear Authors,

the manuscript entitled "Body shape, body composition, and performance on the US Army Combat Fitness Test: Insights from a 3D body image scanner" describes the relationship between body shape and composition with performance in cadets, proposing a new approach mainly on body shape rather than on body composition through evaluation with a new infrared instrument.

1. Overall, it appears a complex and well-structured work with an in-depth data analysis (as detailed in the supplementary sheets). However, in the current version the reader encounters difficulties due to the lack of organization of the text, a fluency of the writing that appears difficult to follow, also given the complexity of the analysis.

The reviewer therefore recommends performing an in-depth analysis in the presentation of the work.

2. In addition, the aim is to demonstrate that body shape has a higher correlation with physical performance than body composition. However, the authors performed DXA only on 47 out of 239 subjects. This represents an major criticism that the authors must justify.

Other minor criticalities are:

-please check some typo as:

in abstract "Some specific anatomical circumferences have higher correlations have higher correlations to performance than body composition."

in discussion "The study17 found that there were significant correlations between fat free mass index (FFMI) and fat free mass index (FFMI) with overall ACFT performance."

- The sentence in What are the new findings?: "The association between body circumferences and performance is stronger than between body circumference and percent body fat." In reviewers' opinion should be add "in this sample of healthy and physically active subjects";

INTRODUCTION:

- please define the abbreviation "RAND";

- in this sentence is not clear if the differences were in the performance or in battery test "Prior to these changes, there were notable differences in performance between males and females on the new United States Army Combat Fitness Test (ACFT)"

- The introduction as a whole appears too general, without a rigorous and aim-oriented orientation of the research.

- It is also advisable to reformulate the purpose of the study, which in the current version is not clear, also in consideration of the use of the BMI parameter which is not the only used in the statistical analysis.

METHODS:

- the sentence does not appear appropriate in the "study design" section, maybe in a "procedures" section: "Relationships between 3D imaged body circumferences, DXA measured body composition, and ACFT performance was examined. While BMI serves as one proxy for body shape, it is desirable to collect more markers of shape through anthropometric measurements 13. However, manually collecting body circumferences can represent a burden to both researcher and participant. Therefore, we used a new technology that obtains measurements quickly and efficiently for analysis. The cadets were scanned using 3D imaging technology which calculated anatomical circumferences at 20 locations on the body. Body composition was also collected with dual energy X-ray " .

- Why the authors performed DXA only a sobgroup of the sample? This should be mentioned in limitations, explaining which statistical methodologies were used to overcome this discrepancy.

- the reviewer suggest to provide a figure for Online Supplemental File 1, instead a text file

RESULTS:

- the overall results section should be simplified, making the reading more fluid.

- in the abstract the following are presented: "The analysis of the clusters produced 5 groups: 1.“ V ”shaped males, 2. older males, 3. inverted“ V ”shaped males and females, 4. males and smaller “V” shaped females, and 5. smaller males and females. ” This description could be used in the results section to guide the presentation of the correlation data obtained by making the chapter more readable.

- Maybe a figure with human body that describe the five cllusters sholud be usefull.

DISCUSSION:

- it is appropriate to hypothesize the reason in the various correlation showed

Reviewer #2: Overall: Interesting article that is definitely value-added to the military. Wonderful job bringing in a variety of tools to examine the relationship between body comp, body shape, and fitness scores, especially given the ACFT is a fairly new requirement. Mine main concern as a reader is keeping the 5 clusters straight and providing a little more detail about the statistical analyses (although I commend you for added the supplementary data). It is very hard for the reader to remember what cluster 2 vs 3 means when considering interpretation of results. One big ask is to develop a 1-2 word descriptor for each cluster, define, and then consistently use throughout. I believe this will transform the manuscript into a much more enjoyable and interpretable manuscript.

Reviewer #3: General comment

- This is an important and well-conducted study. The topic is timely, and this research is sure to make an immediate impact on other researchers (and perhaps application and interpretation of ACFT potential and performance).

- Some fairly minor comments are provided below to strengthen select portions of the manuscript.

Introduction

- The introduction is succinct and includes relevant background information.

Methods

- 3D body image scan measurements section: do you have test-retest reliability data for the circumferences? If not, reliability of Styku-derived anthropometric variables has been published (https://www.nature.com/articles/s41430-019-0526-6) and could be summarized and referenced to provide the reader with context regarding the high reliability of these measurements.

- Body composition section: Similarly, do you have reliability data for the DXA scans? While reliability data would ideally be laboratory-specific, there are, similar to above, other labs that have published reliability data for the model used here (GE Lunar Prodigy) and the variable reported here (body fat %): https://www.cambridge.org/core/journals/british-journal-of-nutrition/article/tracking-changes-in-body-composition-comparison-of-methods-and-influence-of-preassessment-standardisation/D407E50CFAE1CA9297E23CAFE6DC96F7 (see Table 1 for the ICC and TEM). This information would be helpful to the reader to contextualize the error from this measurement.

- K-means cluster analysis section: The phrase “(ratio of the difference of measurement and mean and standard deviation)” is confusing. Could this be rephrased or clarified?

- Were any R packages beyond those included in base R used? If so, those packages should be cited.

Results

- The results presented in-text, through figures and tables, and in the supplementary materials are thorough, informative, and appropriate.

Discussion

- The takeaway point about the provided clusters being more informative than male vs. female clusters is very important, and the authors do a very nice job emphasizing this.

- Please double check and correct if necessary in the second paragraph of the discussion: “The study17 found that there were significant correlations between fat free mass index (FFMI) and fat free mass index (FFMI) with overall ACFT performance.” The term “fat free mass index” seems to be duplicated.

- There is a recent review article (https://pubmed.ncbi.nlm.nih.gov/34593729/) that specified 3D optical scanning in military populations as a future research direction, including the possibility that 3D scanning-derived anthropometrics could be used to inform a recruit’s MOS assignment in conjunction with the Occupational Physical Assessment Test used by the Army. If the authors deem this relevant, examining the section “Potential Applications of a Novel Anthropometric Assessment” within this updated review may provide information to complement and extend the discussion of the application of the findings of the present study.

- Strengths and limitations: similar to a previous comment, providing reliability data – either from your lab or already published – will better support the statement that using the Styku scanner is a strength.

- The authors are commended for identifying the important points about allometric relationships and motivation of cadets during the test.

Supplementary materials

- The authors are commended for the many informative supplemental files they provide. I am certain several of these will be quite helpful to other researchers.

- Please review supplemental file 4 and revise as appropriate. There are 2 pages to the PDF document, but they are very different sizes (the second page is very small and contains a single plot, while the first page is much larger).

- The authors are particularly thanked for providing the R markdown files associated with this analysis.

6. PLOS authors have the option to publish the peer review history of their article (what does this mean?). If published, this will include your full peer review and any attached files.

Reviewer #1: No

Reviewer #2: No

Reviewer #3: No

---

## [Author Response · Author response to Decision Letter 0]

13 Feb 2023

Response to Reviewers

We thank the reviewers for the time they took to go through our manuscript and to suggest changes to improve the manuscript. We took additional time to make all the suggested edits. Some of the concerns were due to flow and structure, so we took time to change the order and describe our process in more detail prior to presenting extensive analysis. We hope these changes make it easier to consume the manuscript. Changes to the text in the manuscript under a specific comment appear in red font

Editorial Review

Comment 1: Please ensure that your manuscript meets PLOS ONE's style requirements, including those for file naming. The PLOS ONE style templates can be found at

Response: We have checked and ensured that our manuscript meets PLoS One’s style requirements.

Comment 2: Thank you for stating the following financial disclosure:

 “The senior author Diana M. Thomas was supported by NIH U54TR004279”

Response: Our NIH grant was split from our co-PI. We now have our own Center and we have changed the grant number accordingly:

“Nicholas Clark, Diana M. Thomas and Nicholas Gist were was supported by National Institutes of Health (NIH) Inter Agency Agreement AOD22022001. The NIH had no role in study design, data collection and analysis, decision to publish, or preparation of the manuscript.”

We have added the statement that funders had no role in study design, data collection and analysis, decision to publish, or preparation of the manuscript.

Comment 3: Please expand the acronym “NIH” (as indicated in your financial disclosure) so that it states the name of your funders in full. Please include this amended Role of Funder statement in your cover letter; we will change the online submission form on your behalf. This information should be included in your cover letter; we will change the online submission form on your behalf.

Response: We have completed as requested.

Comment 4: Thank you for stating the following in your Competing Interests section: “No” Please complete your Competing Interests on the online submission form to state any Competing Interests. If you have no competing interests, please state "The authors have declared that no competing interests exist.", as detailed online in our guide for authors at http://journals.plos.org/plosone/s/submit-now This information should be included in your cover letter; we will change the online submission form on your behalf.

Response: We have now changed this to “The authors have declared that no competing interests exist."

Comment 5: In your Data Availability statement, you have not specified where the minimal data set underlying the results described in your manuscript can be found. PLOS defines a study's minimal data set as the underlying data used to reach the conclusions drawn in the manuscript and any additional data required to replicate the reported study findings in their entirety. All PLOS journals require that the minimal data set be made fully available. For more information about our data policy, please see http://journals.plos.org/plosone/s/data-availability. Upon re-submitting your revised manuscript, please upload your study’s minimal underlying data set as either Supporting Information files or to a stable, public repository and include the relevant URLs, DOIs, or accession numbers within your revised cover letter. For a list of acceptable repositories, please see http://journals.plos.org/plosone/s/data-availability#loc-recommended-repositories. Any potentially identifying patient information must be fully anonymized.

Response: The full data set is available, however, because it is US Military data, a request to use the data needs to be made to the G5 Office of Institutional Research at the United States Military Academy. Upon approval from G5, the data can be found at a link provided by G5. We apologize, but cadets are military personnel and we cannot share the data without authorization from the US Army organization. Important: If there are ethical or legal restrictions to sharing your data publicly, please explain these restrictions in detail. Please see our guidelines for more information on what we consider unacceptable restrictions to publicly sharing data: http://journals.plos.org/plosone/s/data-availability#loc-unacceptable-data-access-restrictions. Note that it is not acceptable for the authors to be the sole named individuals responsible for ensuring data access. We will update your Data Availability statement to reflect the information you provide in your cover letter.

We have now revised Data Availability Statement to read (note Paul Evangelista is not a co-author, but the Chief Data Officer at West Point).

Data will be made available by request after review by the United States Military Academy Chief Data Officer, Paul Evangilista. The review will determine the risks to personnel for data sharing and is dependent on the type of request. Requests can be made at paul.evangelista@westpoint.edu. 

We have also provided this statement in the cover letter:

Due to military security concerns, data on the cadet population at the United States Military Academy cannot be publicly released without a review by the US Military Academy. Data will be made available by request after review by the United States Military Academy Chief Data Officer, Paul Evangelista. The review will determine the risks to personnel for data sharing and is dependent on the type of request. Requests can be made at paul.evangelista@westpoint.edu. Statistical analyses that were performed are available as R Markdown files in the supplementary online information. 

Comment 6: Please include captions for your Supporting Information files at the end of your manuscript, and update any in-text citations to match accordingly. Please see our Supporting Information guidelines for more information: http://journals.plos.org/plosone/s/supporting-information.

Response: We have done this as requested.

7. Please amend the manuscript submission data (via Edit Submission) to include author “Dusty Turner”

Response: We have done this as requested.

Response: We have done this.

Reviewer 1

Comment 1: Overall, it appears a complex and well-structured work with an in-depth data analysis (as detailed in the supplementary sheets). However, in the current version the reader encounters difficulties due to the lack of organization of the text, a fluency of the writing that appears difficult to follow, also given the complexity of the analysis.

The reviewer therefore recommends performing an in-depth analysis in the presentation of the work.

Response: We realized how clunky the manuscript sounded after receiving the reviewer feedback. We re-organized the study analysis in order of why each analysis was performed and now included a study design flow chart as Figure 1. In the study design section, we added the following paragraph (Figure 1 follows):

The study design has three main components. The first is data collection which involved collecting ACFT performance, body shape, and body composition data. The second component focused on correlation between individual body shape parameters and their influence on performance. The final component characterized archetype body shapes identified in the data through a clustering algorithm and then compared performance over these clusters. Figure 1 is a flow diagram that describes the three stages of the study.

Comment 2: In addition, the aim is to demonstrate that body shape has a higher correlation with physical performance than body composition. However, the authors performed DXA only on 47 out of 239 subjects. This represents an major criticism that the authors must justify.

Response: We did not have the personnel power to run DXA scans for all 239 subjects. The maximum number for scanning was 47, however, we selected the participants from the 239 to have this sample be as random as possible. We added the following as a limitation in the discussion:

Another limitation is that we could not measure body composition in all 239 participants. The DXA machine was only available for scanning for an hour a week and only one team member was licensed to conduct the DXA measurements. However, studies that include DXA body composition combined with 3D body image scans are rare41 42. Regardless, future studies should attempt to measure both 3D body image scans simultaneously with DXA measured body composition.

Comment 3: please check some typo as:

in abstract "Some specific anatomical circumferences have higher correlations have higher correlations to performance than body composition."

Response: We have corrected this.

Comment 4: in discussion "The study17 found that there were significant correlations between fat free mass index (FFMI) and fat free mass index (FFMI) with overall ACFT performance."

Response: We have corrected this.

Comment 5: The sentence in What are the new findings?: "The association between body circumferences and performance is stronger than between body circumference and percent body fat." In reviewers' opinion should be add "in this sample of healthy and physically active subjects";

Response: We have made this change.

Comment 6: please define the abbreviation "RAND"

Response: It stands for Research and Development and we have included this expansion of RAND in the manuscript now.

Comment 7: In this sentence is not clear if the differences were in the performance or in battery test "Prior to these changes, there were notable differences in performance between males and females on the new United States Army Combat Fitness Test (ACFT)"

Response: We have clarified it to read:

Prior to these changes, there existed substantial differences in performance between males and females on this event2-4. 

Comment 8: The introduction as a whole appears too general, without a rigorous and aim-oriented orientation of the research.

Response: We have made some changes to the introduction that state this is a discovery science study, not hypothesis testing, and stated the goal of the study. We also expanded the study design section to provide clarity on the study approach and goals. The introduction concludes now with this sentence:

We extended this body of existing work using a discovery science approach to improve understanding of body shape and performance beyond BMI. This was accomplished by examining body shape, body composition, and fitness performance in a United States Military Academy (USMA) cadet population.

Comment 9: It is also advisable to reformulate the purpose of the study, which in the current version is not clear, also in consideration of the use of the BMI parameter which is not the only used in the statistical analysis.

Response: In reading the reviewer comments and reviewing our manuscript, we agree that reading the manuscript is a challenge. We hope the new paragraph in the study design section (see response to Comment 1), the flow diagram (Figure 1), and the section connecting the regression analysis to the cluster analysis combined will help resolve this lack of clarity.

Comment 10: the sentence does not appear appropriate in the "study design" section, maybe in a "procedures" section: "Relationships between 3D imaged body circumferences, DXA measured body composition, and ACFT performance was examined. While BMI serves as one proxy for body shape, it is desirable to collect more markers of shape through anthropometric measurements . However, manually collecting body circumferences can represent a burden to both researcher and participant. Therefore, we used a new technology that obtains measurements quickly and efficiently for analysis. The cadets were scanned using 3D imaging technology which calculated anatomical circumferences at 20 locations on the body. Body composition was also collected with dual energy X-ray.

Response: We agree that the prose does not belong in the study design section. We have removed the first three sentences and replaced with the following:

Data collection utilized new technology that obtained anthropometric measurements quickly and efficiently for analysis. The cadets were scanned using 3D imaging technology which calculated anatomical circumferences at 20 locations on the body. Body composition was also collected with dual energy X-ray absorptiometry (DXA) in a subsample of cadets. These measurements were used to identify different body shape and body composition attributes related to performance on the specific ACFT events.

Comment 11: Why the authors performed DXA only a sobgroup of the sample? This should be mentioned in limitations, explaining which statistical methodologies were used to overcome this discrepancy.

Response: We just didn’t have the personnel and the access to the DXA machine to collect in everyone. We have the following now in the discussion to point out this limitation.

Another limitation is that we could not measure body composition in all 239 participants. The DXA machine was only available for scanning for an hour a week and only one team member was licensed to conduct the DXA measurements. However, studies that include DXA body composition combined with 3D body image scans are rare40 41. Regardless, future studies should attempt to measure both 3D body image scans simultaneously with DXA measured body composition.

Comment 12: The reviewer suggest to provide a figure for Online Supplemental File 1, instead a text file.

Response: A wonderful figure that we could not compete with is Army owned, however, we now reference the Army owned figure in the text. We have supplied the figure below for the reviewer. The change in text now reads:

The six ACFT events15 prior to March 2022 (Online Supplemental File 1), conducted in order with a maximum of 2 minutes between events, include a 3-Repetition Maximum Deadlift (MDL), Standing Power Throw (SPT), Hand-Release Push-Ups (HRPU), Sprint-Drag-Carry (SDC), Leg Tuck (LT), and the 2-mile run16. Graphic descriptions of the current ACFT that replaced LT with a plank can be found in the official ACFT website17. A image description of the LT can be found on the US Army Basic Training website18.

Comment 13: The overall results section should be simplified, making the reading more fluid.

Response: We rearranged the topics and the methods so the topics all follow a clear set of steps. These steps are now conveyed in a Study Design Flow Graphic (see next page). We hope these changes make the results more meaningful, in context, and easier to follow. 

Comment 14: in the abstract the following are presented: "The analysis of the clusters produced 5 groups: 1.“ V ”shaped males, 2. older males, 3. inverted“ V ”shaped males and females, 4. males and smaller “V” shaped females, and 5. smaller males and females. ” This description could be used in the results section to guide the presentation of the correlation data obtained by making the chapter more readable. Maybe a figure with human body that describe the five clusters should be useful.

Response: We love the reviewer’s idea and have developed a graphic with avatars and a short description of the avatars (next page).

Comment 15: It is appropriate to hypothesize the reason in the various correlation showed.

Response: We were unsure if we are being requested to change any portion of the discussion.

Reviewer #2

Comment 1: Mine main concern as a reader is keeping the 5 clusters straight and providing a little more detail about the statistical analyses (although I commend you for added the supplementary data). It is very hard for the reader to remember what cluster 2 vs 3 means when considering interpretation of results. One big ask is to develop a 1-2 word descriptor for each cluster, define, and then consistently use throughout. I believe this will transform the manuscript into a much more enjoyable and interpretable manuscript. 

Response: We have now included a figure with avatars in each cluster with descriptions. We also included body shape types (“V” shaped etc.). The figure is pasted on the next page for the Reviewer’s convenience.

Reviewer #3

Comment 1: 3D body image scan measurements section: do you have test-retest reliability data for the circumferences? If not, reliability of Styku-derived anthropometric variables has been published (https://www.nature.com/articles/s41430-019-0526-6) and could be summarized and referenced to provide the reader with context regarding the high reliability of these measurements.

Response: I’m not aware that there is a test-retest reliability study but we have seen the Styku article and have now included this as a reference. The change is pasted below.

Between February and March of 2021, participants arrived at a designated location to be scanned using the Styku 3D body imaging machine (Styku S100, Advanced Model of Styku Phoenix software; Los Angeles, CA). Participants stood on a rotating turntable with their arms held in an “A” pose. As the turntable revolved for 35 seconds, a tower 114 cm away captured images using an infrared camera equipped with “Time of Flight” technology that quantifies the depth in an image19. Participants wore either their USMA issued swimsuit or compression shorts with a sports bra for female participants. The Styku internal software generates body circumference measurements at 20 different locations, ratios of chest-to-waist, chest-to-hip and waist-to-hip, and body weight. Styku has been validated against manual measurements for reliability20. All statistical analysis used averaged left and right limb circumferences (Figure 1), which reduced to 13 averaged circumferences. 

Comment 2: Body composition section: Similarly, do you have reliability data for the DXA scans? While reliability data would ideally be laboratory-specific, there are, similar to above, other labs that have published reliability data for the model used here (GE Lunar Prodigy) and the variable reported here (body fat %): https://www.cambridge.org/core/journals/british-journal-of-nutrition/article/tracking-changes-in-body-composition-comparison-of-methods-and-influence-of-preassessment-standardisation/D407E50CFAE1CA9297E23CAFE6DC96F7 (see Table 1 for the ICC and TEM). This information would be helpful to the reader to contextualize the error from this measurement.

Response: We have now included the suggested reference on reliability data for the GE Lunar Prodigy. The change in text appears below:

Body composition was measured by DXA (General Electric Healthcare, Lunar Prodigy Advance, Madison, WI) in 47 participants who were already scanned with the Styku. Similar to the Styku, the Lunar Prodigy has published reliability data 21. Participants recruited for DXA scans reported to the human performance laboratory and were scanned by the same investigator during the months between April and May 2021.

Comment 3: K-means cluster analysis section: The phrase “(ratio of the difference of measurement and mean and standard deviation)” is confusing. Could this be rephrased or clarified?

Response: We spilt the sentences to make this more clear. The change in text appears below:

To group the participant population by similar body shapes, a k-means cluster analysis was performed over the normalized body shape measurements. The normalized body shape measurements of the 13 circumference sites, body weight, and body height were computed by taking the difference of the measurement and the mean value. This difference was then divided by the standard deviation.

Comment 4: Were any R packages beyond those included in base R used? If so, those packages should be cited.

We used tidyr. Apparently the kmeans command is part of base R. We have now cited tidyr:

All statistical analyses were performed in the software package R (R Core Team (2013)). The R package “tidyr”13 was used to perform summary statistics by grouping. 

Comment 5: Please double check and correct if necessary in the second paragraph of the discussion: “The study17 found that there were significant correlations between fat free mass index (FFMI) and fat free mass index (FFMI) with overall ACFT performance.” The term “fat free mass index” seems to be duplicated.

Response: Yes it was duplicated and should have been FFMI and FMI for fat mass index. We have corrected this.

Comment 6: There is a recent review article (https://pubmed.ncbi.nlm.nih.gov/34593729/) that specified 3D optical scanning in military populations as a future research direction, including the possibility that 3D scanning-derived anthropometrics could be used to inform a recruit’s MOS assignment in conjunction with the Occupational Physical Assessment Test used by the Army. If the authors deem this relevant, examining the section “Potential Applications of a Novel Anthropometric Assessment” within this updated review may provide information to complement and extend the discussion of the application of the findings of the present study.

Response: This was an excellent find and we were not tracking. 

Comment 7: Strengths and limitations: similar to a previous comment, providing reliability data – either from your lab or already published – will better support the statement that using the Styku scanner is a strength.

Response: We agree and added text in the discussion:

The use of the Styku 3D scanner, was a major strength of the study. The Styku has undergone validity and reliability testing compared to manual measurements21. Using the Styku allowed for numerous anthropometry measurements collected quickly and reliably in the cadet population at USMA. In addition, the carefully curated routine collection and storage of physical performance data at USMA allowed us to examine relationships between body shape and ACFT performance. 

Comment 8: Please review supplemental file 4 and revise as appropriate. There are 2 pages to the PDF document, but they are very different sizes (the second page is very small and contains a single plot, while the first page is much larger).

Response: We agree with the reviewer, however, this is the editorial manager uploaded change that resulted when the file was merged in. This is not what it looks like as an individual file. We hope that each file will be available separately on the PloS One site.

Reviewer 3 Comments that were inserted in the PDF

Most of Reviewer 3’s comments were cosmetic text changes that improved the reading. We have made all the suggested changes of this nature. 

The following were methodological comments which we address point by point:

Comment 1: Sample size to compare so many variables via correlation plots.

Response: We are not performing multiple comparisons so a Bonferroni type correction was not necessary, however, the Reviewer brings up a good point on whether the sample is large enough to justify statistical inference. We have performed a sample size calculation to justify that the 239 sample was large enough to make the inferences we have. We added the following section:

Sample size estimates

With significance level, α=0.05 (two-tailed), 80% power and an expected correlation of r=0.20, the total sample size required to determine whether a correlation coefficient differs from zero was 19413. For multiple regression, we applied Green’s rule14 for 13 predictors. Setting significance level, α=0.05 (two-tailed), a medium effect (Cohen’s f=0.39), and 80% power, we arrived at a sample size of 154 to test the entire model and 117 to test the coefficients. Our convenience sample of 239 satisfied the sample size requirements.

Comment 2: On the line: “Six different linear regression models were developed using all circumferences and ratios as independent variables and event performance as the dependent variable.” Vague. can you provide a little more information? Assuming multiple linear regression since you had a continuous dependent variable and more than 2 indep variables. Did you use forward or backward stepwise analysis?

Response: We agree with the reviewer and added more description to this section. We have pasted the added text below for the reviewer’s convenience. We also believe that the description multiple regression did not fit well with the flow as noted by the other reviewers. 

We did use forward stepwise analysis but didn’t include it in the publication. This is because, the main point of the regression is to 1) see which variables impacted predicted performance the most and 2) does clustering help us understand performance. The standardized coefficients addresses 1) but 2) was missing. To address 2) we now included regression models that include cluster number as a co-variate to see how well membership in the cluster predicts performance. We have run the analysis for 2) and now included a section in the methods and the results describing the results which we hope the reviewer finds interesting. These sections are pasted below for the reviewer’s convenience. 

We also think that the flow of the analysis was clunky (noted by the other reviewers). We have re-organized the topics so the flow is improved and included a study design flow chart. We have pasted this at the end for the reviewer.

Revision for description of multiple linear regression models.

Multiple linear regression models were developed using the 13 circumferences and ratios as independent variables and event performance as the dependent variable. Specifically, an individual multiple linear regression model was developed to predict ACFT performance scores; MDL, SPT, HRPU, SDC, LT, and the 2-mile run15 from the Styku generated measurements.

Regression models with Cluster membership as a co-variate:

In the Methods Section:

Regression Models with Cluster Number as a covariate

To see if cluster assignment improves prediction of performance on the ACFT events, we developed 6 multiple linear regression models with two independent variables, cluster number and sex. The explanatory variable was the performance score of the ACFT event.

In the Results section:

Regression Models with Cluster Number as a covariate

Similar adjusted R2 were found for the six models predicting performance on each ACFT event compared to the original multiple linear regression models that were developed without cluster number as a covariate. The full statistical output can be found in Online Supplemental File 7 as a R Markdown file.

Comment 3: The reviewer remarked that the cluster types were hard to follow. Other reviewers noted the same and we have now included the figure to help visualize the cluster members.

---

## [Decision Letter · Decision Letter 1]

2 Mar 2023

PONE-D-22-21891R1Body shape, body composition, and performance on the US Army Combat Fitness Test: Insights from a 3D body image scannerPLOS ONE

Dear Dr. Thomas,

Thank you for submitting your manuscript to PLOS ONE. After careful consideration, we feel that it has merit but does not fully meet PLOS ONE’s publication criteria as it currently stands. Therefore, we invite you to submit a revised version of the manuscript that addresses the points raised during the review process.

Two minor, albeit correct remarks from the reviewers:

- The body composition part is not fully within the scope of the study and given the lack of data it is suggested that the DXA analysis part (also from the title) be deleted. This would make the manuscript more organized and linear, with a clearer purpose of the study. Given the sample size, it seems to be a solution that allows for better understanding of the text.

- The purpose of the study is not clear at the end of the introduction section. Please submit your revised manuscript by Apr 16 2023 11:59PM. If you will need more time than this to complete your revisions, please reply to this message or contact the journal office at plosone@plos.org. Please include the following items when submitting your revised manuscript:A rebuttal letter that responds to each point raised by the academic editor and reviewer(s). You should upload this letter as a separate file labeled 'Response to Reviewers'.A marked-up copy of your manuscript that highlights changes made to the original version. You should upload this as a separate file labeled 'Revised Manuscript with Track Changes'.An unmarked version of your revised paper without tracked changes. You should upload this as a separate file labeled 'Manuscript'.If applicable, we recommend that you deposit your laboratory protocols in protocols.io to enhance the reproducibility of your results. Protocols.io assigns your protocol its own identifier (DOI) so that it can be cited independently in the future. For instructions see: https://journals.plos.org/plosone/s/submission-guidelines#loc-laboratory-protocols. Additionally, PLOS ONE offers an option for publishing peer-reviewed Lab Protocol articles, which describe protocols hosted on protocols.io. Read more information on sharing protocols at https://plos.org/protocols?utm_medium=editorial-email&utm_source=authorletters&utm_campaign=protocols.

We look forward to receiving your revised manuscript.

Kind regards,

Inge Roggen, M.D., Ph.D.

Academic Editor

PLOS ONE

Journal Requirements:

Reviewers' comments:

Reviewer's Responses to Questions

**Comments to the Author**

1. If the authors have adequately addressed your comments raised in a previous round of review and you feel that this manuscript is now acceptable for publication, you may indicate that here to bypass the “Comments to the Author” section, enter your conflict of interest statement in the “Confidential to Editor” section, and submit your "Accept" recommendation.

Reviewer #1: All comments have been addressed

Reviewer #3: All comments have been addressed

2. Is the manuscript technically sound, and do the data support the conclusions?

Reviewer #1: Partly

Reviewer #3: Yes

3. Has the statistical analysis been performed appropriately and rigorously? 

Reviewer #1: Yes

Reviewer #3: Yes

4. Have the authors made all data underlying the findings in their manuscript fully available?

Reviewer #1: (No Response)

Reviewer #3: Yes

5. Is the manuscript presented in an intelligible fashion and written in standard English?

Reviewer #1: (No Response)

Reviewer #3: Yes

6. Review Comments to the Author

Reviewer #1: Dear Authors

the revised version of the manuscript appear improved.

However, in the reviewer's opinion, the body composition part is not fully within the scope of the study and given the lack of data it is suggested that the DXA analysis part (also from the title) be deleted. This would make the manuscript more organized and linear, with a clearer purpose of the study. Given the sample size, it seems to be a solution that allows for better understanding of the text.

Specific comment:

- The purpose of the study is not clear at the end of the introduction section.

Reviewer #3: Thank you for addressing the reviewer comments. This is an important manuscript that will interest many readers.

7. PLOS authors have the option to publish the peer review history of their article (what does this mean?). If published, this will include your full peer review and any attached files.

Reviewer #1: No

Reviewer #3: No

---

## [Author Response · Author response to Decision Letter 1]

8 Mar 2023

Thank you to the reviewers for the time and effort they took to improve the manuscript. We realized that a previously reviewed section that had no comments was accidently omitted in Revision 2 when we rearranged the flow of the manuscript. In addition a section that appeared in the response to reviewers on regression models with cluster as a independent variable was omitted from the manuscript We have placed these sections back at the end of the results (highlighted) and paste it below here for the Reviewer’s convenience. 

ACFT performance by cluster

Overall average performance on every ACFT event was highest in Cluster 1 on every event except the SPT (Table 1, Box Plots Online Supplemental File 3). Participants classified to Cluster 2 had the highest performance on the SPT event and second highest performance on the MDL and SDC events. Individuals in this cluster had similar performance to Cluster 4 on the HRPU event and performed below Cluster 4 on the LT and the 2-mile run. Participants in Cluster 3 performed above Cluster 5 on all events except the 2-mile run. 

Comparison of means between clusters revealed that Clusters 1 and 2 had statistically significant better performance than Clusters 3, 4 and 5 for the MDL (Online Supplemental File 5). Additionally, Clusters 3 and 4 had significantly better performance on the MDL compared to Cluster 5. For SPT, HRPU and SDC and LT, Clusters 1 and 2 had statistically significant better performance than Clusters 3 and 5. Finally, Cluster 4 had statistically significant better performance on the SDC and LT compared to Cluster 5. There were no statistically significant differences in performance over any ACFT event between Cluster 3 and Cluster 4.

Regression Models with Cluster Number as a covariate

Similar adjusted R2 were found for the six models predicting performance on each ACFT event compared to the original multiple linear regression models that were developed without cluster number as a covariate. The full statistical output can be found in Online Supplemental File 6 as a R Markdown file.

We had only two comments to address which we address below:

Reviewer #1

Comment 1: Dear Authors the revised version of the manuscript appear improved.

However, in the reviewer's opinion, the body composition part is not fully within the scope of the study and given the lack of data it is suggested that the DXA analysis part (also from the title) be deleted. This would make the manuscript more organized and linear, with a clearer purpose of the study. Given the sample size, it seems to be a solution that allows for better understanding of the text.

Response: We have removed all references to body composition, percent fat and DXA in the manuscript. We have eliminated body composition from the title. These changes are not visible in the marked up manuscript, because you cannot highlight a section you have deleted, but we hope the reviewer can see the change easily.

Comment 2: The purpose of the study is not clear at the end of the introduction section.

Response: We agree and we have changed this last paragraph in the introduction to reflect the purpose of the study. The paragraph is pasted below for the reviewer’s convenience.

We wanted to understand the role body shape on ACFT performance. Specifically, what are the archetype body shapes that predispose individuals to better ACFT performance and alternatively which archetype body shapes that are associated to poorer ACFT performance. Addressing these questions can offer opportunities for training programs to target specific body sites associated with better ACFT performance.

---

## [Editor Report · Decision Letter 2]

13 Mar 2023

Body shape and performance on the US Army Combat Fitness Test: Insights from a 3D body image scanner

PONE-D-22-21891R2

Dear Dr. Thomas,

We’re pleased to inform you that your manuscript has been judged scientifically suitable for publication and will be formally accepted for publication once it meets all outstanding technical requirements.

Kind regards,

Inge Roggen, M.D., Ph.D.

Academic Editor

PLOS ONE
---

## [Editor Report · Acceptance letter]

6 Apr 2023

PONE-D-22-21891R2 

Body shape and performance on the US Army Combat Fitness Test: Insights from a 3D body image scanner 

Dear Dr. Thomas:

I'm pleased to inform you that your manuscript has been deemed suitable for publication in PLOS ONE. Congratulations! Your manuscript is now with our production department. 

Kind regards, 

on behalf of

Prof. Inge Roggen 

Academic Editor

PLOS ONE